# Direct observation of desorption of a melt of long polymer chains

Xavier Monnier [1,4], Simone Napolitano [2,4✉] & Daniele Cangialosi [1,3,4✉]

Tuning the thermodynamic state of a material has a tremendous impact on its performance. In the case of polymers placed in proximity of a solid wall, this is possible by annealing above the glass transition temperature, $T_g$, which induces the formation of an adsorbed layer. Whether heating to higher temperatures would result in desorption, thereby reverting the thermodynamic state of the interface, has so far remained elusive, due to the interference of degradation. Here, we employ fast scanning calorimetry, allowing to investigate the thermodynamics of the interface while heating at $10^4$ K s$^{-1}$. We show that applying such rate to adsorbed polymer layers permits avoiding degradation and, therefore, we provide clear-cut evidence of desorption of a polymer melt. We found that the enthalpy and temperature of desorption are independent of the annealing temperature, which, in analogy to crystallization/melting, indicates that adsorption/desorption is a first order thermodynamic transition.

[1] Donostia International Physics Center, Paseo Manuel de Lardizabal 4, 20018 San Sebastián, Spain. [2] Laboratory of Polymer and Soft Matter Dynamics, Experimental Soft Matter and Thermal Physics (EST), Faculté des Sciences, Université libre de Bruxelles (ULB), CP223, Boulevard du Triomphe, 1050 Brussels, Belgium. [3] Centro de Física de Materiales (CSIC–UPV/EHU), Paseo Manuel de Lardizabal 5, 20018 San Sebastián, Spain. ✉email: snapolit@ulb.ac.be; daniele.cangialosi@ehu.eus

Adsorption is a ubiquitous phenomenon affecting the properties of all types of interfaces[1–3]. While approaching an adsorbing wall, particles lose their translational and rotational freedom and rearrange by adopting spatial configurations differing from those far from the interface[4]. The related changes in particle density, at the boundary between adsorbent and adsorbate, determine the performance of the whole adsorbed layer[5]. The work of adhesion—a key parameter for several engineering applications—is, in fact, directly proportional to the number of contacts formed by two neighboring media, and consequently to the amount of matter adsorbed at the interface[1,6].

In the last decade, significant interest has grown around adsorbed polymer layers[7], because of a series of striking correlations identified between the number of chains adsorbed and a broad ensemble of materials properties. Glass transition temperature[8,9], viscosity[10], mechanical properties[11], lateral diffusion[12], crystallization rate[13], and maximum water uptake[14], are just a few examples of the quantities influenced by the adsorbed amount. Being able to control the latter quantity (e.g., by simple thermal treatments applicable at the industrial scale) would open the way to new processing strategies to finely tune the interfacial density[7,15] and, hence, the properties of nanomaterials.

Remarkably, polymers adsorb more easily than small molecules. Adsorption of polymers can, in fact, take place already at monomer/substrate interactions smaller than $k_BT$[16,17]. In such case, while adsorption is still reversible at the monomer level, desorption of a whole chain is less likely to occur, because it requires simultaneous detachment of a large number of adsorbed monomers[18,19]. As the probability that such an event would occur within the timescales and at the temperatures of technological interest is very low—that is, the energetic barrier for desorption is very large—, for practical purposes, chain adsorption is considered as irreversible. This reasoning is, however, merely based on kinetics. The adsorbed amount (or equivalently the thickness of an adsorbed layer[7]) is, in fact, an equilibrium quantity related to the amount of free chains present in the environment, e.g., the concentration of a polymer solution. The number of adsorbed chains increases with polymer concentration until reaching a saturation value, given by interfacial interactions. Desorption of adsorbed chains is, hence, possible by placing a polymer layer in a pure solvent[20], or in a solution whose concentration is lower than that related to the equilibrium adsorbed amount[4]. Importantly,

the concentration thresholds to stabilize adsorbed layers are rather low, which corresponds to weak desorption forces[21,22], and extremely long desorption times: soaking for 4 months in pure chloroform is not sufficient to fully desorb an adsorbed layer of polystyrene (PS)[17].

The physical picture drawn by these studies is that the process of adsorption is thermodynamically driven and, therefore, this phenomenon can be depicted as a phase transition of a non-adsorbed "bulk" polymer melt into a tightly adsorbed layer. This transformation entails a reduction of the enthalpy and, to a smaller extent, also of the entropy. As a consequence, with increasing temperature the larger weight of entropic effects is expected to equalize the free energy of the two phases, thereby reverting the transformation and the opposite phenomenon, that is, desorption of polymer chains from the solid substrate, should be observed. At the temperature at which desorption occurs, a phase transition is, therefore, expected. Though way less studied and poorly understood[23], such phenomenon is as relevant as that of adsorption, because it allows varying interfacial properties. In spite of this, a signature of desorption of a polymer melt from a solid substrate has so far remained unidentified. The main reason is that polymers heated at standard rates (~1–10 K min$^{-1}$) generally undergo degradation before desorption.

To overcome this problem, here we employ fast scanning calorimetry (FSC), which permitted us to determine the calorimetric response of thin polymer layers processed at $10^4$ K s$^{-1}$, thereby avoiding degradation. We provide compelling evidence for the desorption of a previously adsorbed layer, as revealed by an endothermic overshoot at high temperatures, much higher than the polymer glass transition temperature ($T_g$). In analogy with the response observed for crystallization/melting processes, the occurrence of this peak indicates the presence of a first-order thermodynamic transition, in line with previous theoretical work.

## Results

**Calorimetric response of thin films upon fast heating.** To design our experiments, we considered the expected response of an interfacial polymer layer upon heating from both kinetic and thermodynamic viewpoints (Fig. 1). The temperature dependence of free energy is schematically reported in Fig. 1a, whereas the role of heating rate is explicitly considered in the

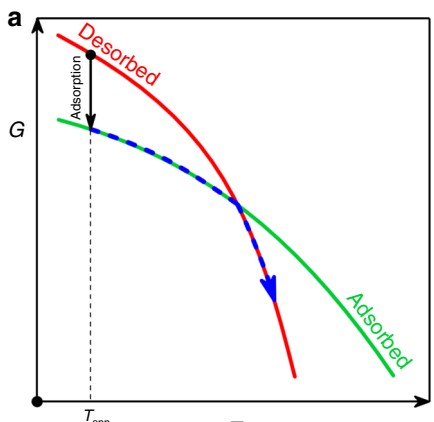
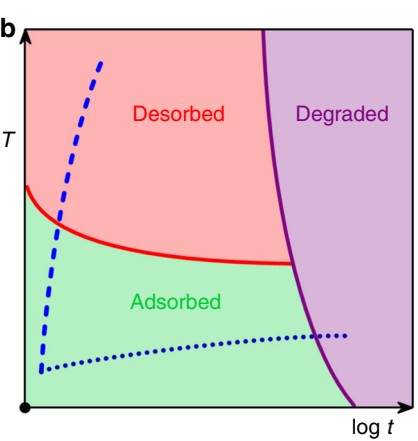

**Fig. 1 Phase diagram and kinetic stability of an adsorbed layer. a** Schematic representation of the dependence of Gibbs-free energy ($G$) on temperature ($T$) for the desorbed (red line) and the adsorbed (green line) phases. In our experiments, adsorption is achieved isothermally at low temperatures; the adsorbed layers are then heated at a constant rate up to high temperatures where desorption takes place spontaneously. The blue dashed line indicates the thermodynamic path followed by heating the sample at a high rate. **b** Schematic representation of TTT (time–temperature–transformation) diagram for an adsorbed polymer layer. The blue dashed line and the dark blue dotted line indicate CHT (constant heat transformation) processing protocols at, respectively, fast and slow rate. Degradation is circumvented by heating the adsorbed layer at a fast scanning rate.

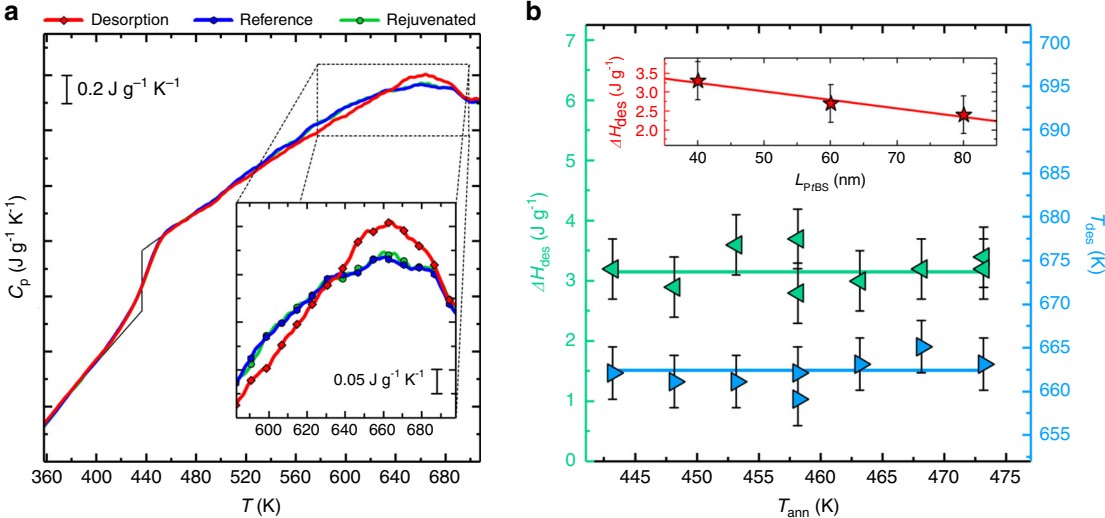

**Fig. 2 Desorption transition in PtBS thin films. a** Heating scans at $10^4$ K s$^{-1}$ after annealing at 458 K for $5 \times 10^4$ s (red curves), before adsorption (reference, blue curves) and after desorption (rejuvenated, green curves). Black lines at about 440 K highlight the step in the specific heat associated with the glass transition. The inset of **a** is an enlargement in the temperature range where desorption takes place. **b** Enthalpy of desorption (left triangles) and temperature of the maximum of the endothermic overshoot (right triangles) as a function of the annealing temperature. The inset of panel **b** shows the enthalpy of desorption after annealing for $5 \times 10^4$ s at 458 K as a function of the film thickness. Error bars account for the uncertainty in the determination of the temperature maximum ($T_{des}$) and intensity ($\Delta H_{des}$) of the overshoot.

time–temperature–transformation (TTT) diagram pictured in Fig. 1b. We used thin films of poly(4-*tert*-butylstyrene) (PtBS), a polymer whose glassy dynamics[24], vitrification[25], and adsorption kinetics[9] have been deeply characterized in the previous years. We prepared our samples by spin-coating, the most commonly used method to coat solid substrates with thin polymer layers. Solutions of PtBS were directly deposited onto the backside of the calorimetric chips, which consist of a thick layer of silicon oxide. By removing non-adsorbed chains via conventional methods[16], we checked that no continuous adsorbed layer is formed in as spin-coated samples.

We begin our discussion presenting the calorimetric response of 60-nm-thick PtBS films annealed at 458 K during $5 \times 10^4$ s— these conditions are sufficient to ensure adsorption of polymer chains onto the substrate[9,16]. The outcome of these experiments is shown in Fig. 2a. At about 440 K, PtBS undergoes the glass transition, as indicated by a step in the specific heat capacity. Apart from this, all heating scans successive to adsorption show another common feature. A clear peak in the heat capacity, $C_p$, appearing as an endothermic overshoot with respect to the reference scan obtained prior to adsorption, is located at ~660 K. This overshoot, corresponding to an increase of the enthalpy of the melt, is not present in further heating scans, labeled as "rejuvenated", where the calorimetric response overlaps with the reference scan. This experimental evidence indicates that after heating to 743 K, PtBS thin layers respond as pristine non-adsorbed films. Based on these results, we remark that the endothermic overshoot is observed only in the presence of PtBS chains adsorbed onto the chip. Therefore, such a peak is straightforwardly interpreted as a signature of desorption of the polymer melt from the solid substrate. A quantification of the amount of enthalpy of desorption, obtained by integrating the area between the endothermic overshoot and the respective reference/rejuvenated scans (Supplementary Fig. 1), is provided in Fig. 2b. Importantly, the heat of desorption does not depend on the annealing temperature (the outcome of experiments performed at other annealing temperatures is given in Supplementary Fig. 3a–c). This result indicates that annealing at different temperatures always delivers the same adsorbed amount,

and that such amount is entirely desorbed within our experimental conditions. The enthalpy of desorption is about 3 J $m_{sample}^{-1}$, where $m$ is the mass, which—considering that the thickness of the adsorbed layer is of the order of 5 nm[9,16], that is, about 10 % of the whole sample—corresponds to an enthalpy of adsorption normalized to the mass of the adsorbed layer of ~30 J g$^{-1}$. Similarly to the enthalpy, the temperature of desorption, taken as the maximum of the endothermic overshoot, is also invariant with the annealing temperature as displayed in Fig. 2b. Remarkably, when varied over a range between 40 and 80 nm, the film thickness does not have any role in determining the final amount of enthalpy and the temperature range of desorption: the scaling in the inset of Fig. 2b confirms that the heat exchanged by the sample is thickness independent, as shown in detail in Supplementary Fig. 3 and further commented in Supplementary Note 1.

Experiments on thin films of PS confirmed the trend observed for PtBS. First, the calorimetric response of samples annealed for a time sufficient to form an adsorbed layer presents a peak in the heat capacity at temperatures well above the glass transition. Second, such a feature is not present in rejuvenated samples and, furthermore, the annealing temperature used for adsorption experiments does not affect the heat of desorption nor its temperature range. See Supplementary Fig. 4 and Supplementary Note 2 for desorption experiments on PS thin films.

**Accessing adsorption kinetics via desorption experiments**. To further test the robustness of our approach, we performed another set of experiments, where we monitored the kinetics of adsorption. Specifically, we annealed non-adsorbed films of PtBS for different times at different temperatures. Figure 3 shows the time evolution of the endothermic overshoot due to desorption after annealing for different times at 458 K. In such conditions, the adsorbed amount increases upon annealing[9]. Because of the linear correlation between the number of adsorbed chains and heat released upon desorption, the progressive increase of the desorption enthalpy with annealing time indicates a corresponding growth of the number of chains irreversibly adsorbed

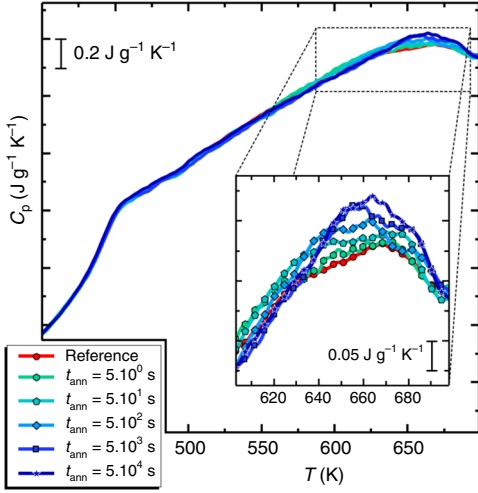

**Fig. 3 Dependence of the heat of desorption on annealing.** Heating scans at $10^4$ K s$^{-1}$ after annealing at 458 K for various times. The inset is an enlargement in the temperature range where desorption takes place.

onto the substrate. In the previous set of experiments, we have remarked that heating the sample up to 743 K, that is, at temperatures higher than the offset of the desorption peak, induces desorption of all the adsorbed chains. Given this observation, the evolution of the enthalpic overshoot with annealing time is a measurement of the kinetics of adsorption. We performed this type of experiment at three annealing temperatures. The outcome of this investigation is shown in Fig. 3. Regardless of the temperature of annealing, the enthalpy of desorption increases following a logarithmic growth in time, typical of the kinetics of adsorption[7,16,26], till reaching 3.2 J g$^{-1}$. However, the kinetics is particularly sensitive to the annealing conditions and it slows down with decreasing temperature, which is typical of thermally activated processes. To determine the molecular origin of the enthalpy increase, we considered the equilibration time, $\tau_{eq}$, to reach the plateau. Values of $\tau_{eq}$ as a function of the inverse temperature are shown in the inset of Fig. 4.

## Discussion

The characterization of desorption, similar to other processes occurring in the soft matter well above room temperature, is limited by the occurrence of degradation phenomena. At an industrial level, this problem is circumvented, because fast processing permits avoiding exposure of the material at high temperatures for a time longer than that necessary to start degradation. Optimization of these processes, currently proceeding via lengthy inefficient hit-and-try procedures, pushed toward the development of experimental methods capable to measure within very short acquisition times. Our innovative study on desorption fits within this scope. Thanks to FSC, a technique capable to determine within 0.1 ms heat flow rate exchanges as small as 0.01 mW between sample and environment[27,28], we could observe desorption before degradation occurred. Moreover, the desorption temperature measured in our experiments (peak maximum at ~660 K) is within the range of previous works, where FSC was successfully used to investigate high-temperature processes as the melting of silk (560 K)[29], the glass transition of anhydrous starch polymers (590 K)[30], and that of high-performance gas-separation membranes (715 K)[31,32].

The impact of our work goes by far beyond the mere development of the state of the art of fast scanning calorimetry. The phenomenology associated with adsorption/desorption is of utmost importance in nanoconfined polymer materials, where a

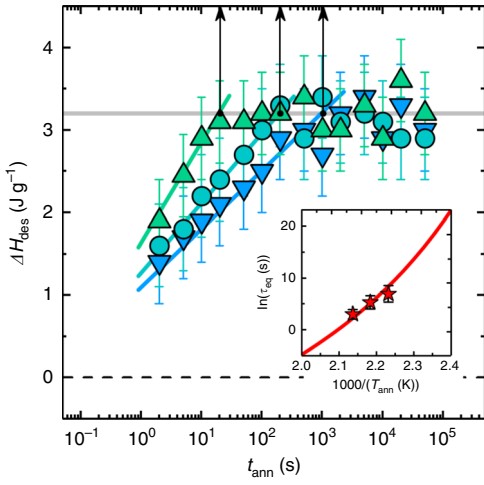

**Fig. 4 Evolution of the enthalpy of desorption with annealing time.** The heat of desorption increases with annealing time, faster kinetics are observed at higher temperatures (448 K blue down-pointed triangles, 458 K cyan circles, 468 K green up-pointed triangles). Vertical arrows indicate the values of $\tau_{eq}$, the timescale to reach a plateau in the enthalpy of desorption, at each temperature. Error bars account for the uncertainty in the determination of the intensity of the overshoot. In the inset, red stars indicate the values of $\tau_{eq}$, whereas the red line is obtained from the temperature dependence of the segmental mobility, taken from ref. [24]. Error bars account for the uncertainty in the determination of the time necessary to reach a constant $\Delta H_{des}$ value in the data set shown in the main panel of the same figure.

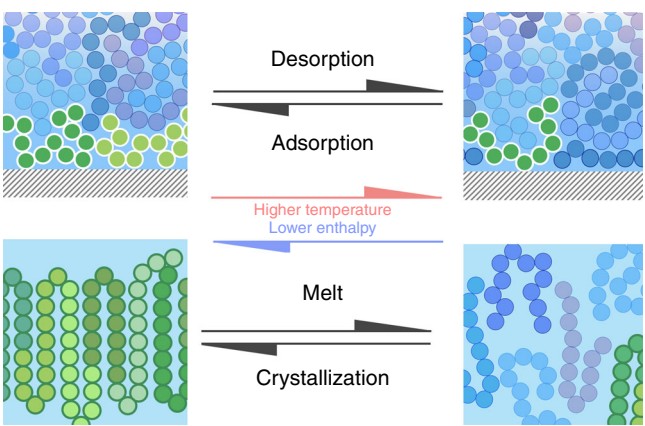

**Fig. 5 Adsorption/desorption vs. crystallization/melting.** In the schematics, adsorbed and crystallized chains are given in tones of green, whereas other chains are in tones of blue. Increasing the temperature of an interfacial polymer layer induces a phase transition related to a sudden reduction in the number of macromolecules adsorbed per unit surface. This process has a high affinity with the melting of a polymer crystal.

large amount of interface exists. Therefore, its relevance can be viewed from both fundamental and technological perspectives, given the increasing importance of miniaturization in diverse applications. At the same time, an analogy with crystallization/melting, a well-known phenomenon in condensed matter physics, can be drawn. This analogy is schematized in Fig. 5. Both crystallization/melting and adsorption/desorption entail a first-order thermodynamic transformation. This implies a reduction of enthalpy and entropy in both adsorption and crystallization and the opposite in desorption and melting. With this analogy in mind, in our work, we show how adsorption kinetics can be followed in ways similar to crystallization kinetics, where the

subsequent heat of desorption quantifies the amount of previously adsorbed polymer.

An interesting analysis regards the temperature of the phase transformation, that is, the temperature at which the free energy of the adsorbed state equals that of the desorbed state. This corresponds to the maximum of the endothermic overshoot ($\approx$663 K), which is well above the polymer $T_g$, at ~398 K. The large value of the desorption temperature can be understood considering that both adsorbed and desorbed chains are in the disordered state and, therefore, their entropy difference is relatively small. As a result, compensation of enthalpic effects and the equalization of free energies require considerably higher temperatures. Indeed, the entropy change can be easily calculated as $\Delta S_{des} = \Delta H_{des}/T_{des}$, which from a measurement of the enthalpy and temperature of desorption, $\approx$30 J g$^{-1}$ = 4800 J mol$^{-1}$ and 663 K, respectively, provides an entropic contribution of ~7.3 J mol$^{-1}$ K. This quantifies the marginal role of entropy in the transition adsorption/desorption considering that, for instance, the entropy associated to the transition between two states is $k \ln 2 = 5.8$ J mol$^{-1}$ K. From the above discussion, we point out that the need of reaching high temperatures implies that fast scanning rates are an essential pre-requisite to observe polymer desorption.

Our results allow attaining key information regarding the origin of the enthalpy variation with adsorption/desorption. In particular, this can come from two contributions: the energy exchange on pinning/unpinning monomers at the polymer/substrate interface and the conformational modifications underlying the liquid–liquid transition between adsorbed and desorbed polymer. The former can be estimated considering the energy of each pinpoint. For polymers interacting with substrates via van der Waals forces, this is of the order of 1 $k_BT$[1,33], which at the temperatures at which adsorption/desorption takes place for P$t$BS corresponds to ~10$^{-20}$ J. Considering that no more than one Kuhn segment can be adsorbed on a surface on the order of 1 nm$^2$, we get an upper bound of 1 pinpoint per nm$^2$, which implies energy exchanged per unit area of ~10$^{-20}$ J nm$^{-2}$. Given the density, $\rho$, of P$t$BS (0.95 g cm$^{-3}$) and the total film thickness of the sample, $h$, the exchanged enthalpy $\Delta H_{pinpoints} = E/\rho h \approx 0.18$ J g$^{-1}$. Importantly, this contribution is limited to only 6% of the total enthalpy exchanged (3.2 J g$^{-1}$, Fig. 4). This result indicates that the vast majority of the enthalpy exchange resulting from adsorption/desorption transition must be attributed to a variation in conformational energy, that is, a perturbation in density.

Another important aspect of our study is related to the difference between our experimental procedure, where a polymer melt is desorbed by fast annealing well above $T_g$, and previous work where a reduction in adsorbed amount was obtained in artificial conditions. Desorption was, for example, studied by displacing adsorbed chains with molecules having a larger affinity with the substrate[18,22,33–35], or by varying the quality or the volume fraction of the solvent[17,20]. These experiments require the introduction of different types of molecules in the system, and hence, the polymer chains undergoing desorption are submitted to thermodynamic conditions way far from those of a pure polymer melt. The resulting kinetics of desorption is, consequently, strongly conditioned by the chemical environment. Fast-controlled heating permitted to avoid these limitations and allowed us to investigate the genuine response of a polymer melt.

Moreover, the methodology presented here allows attaining insights on the molecular mechanisms involved in the kinetics of adsorption. The experimental protocol described in Fig. 3 opens to the possibility of investigating the kinetics of adsorption of polymer films via calorimetry. The increase in adsorbed amount was followed by holding the samples for a given time at the

adsorption temperature and then measuring the heat exchanged upon desorption. The presence of the desorption peak can be uniquely attributed to previously adsorbed polymer chains, given that the subsequent heating scan perfectly overlaps with the reference scan. We remark that the adsorption kinetics could not be directly monitored by measuring the heat exchanged in real-time, because this quantity is several orders of magnitude smaller than the instrument resolution. Based on the analysis of the temperature dependence of the kinetics in Fig. 4, we highlight two main points. First, regardless of the temperature, the enthalpy of desorption saturates to the same value, which implies that the adsorbed amount is temperature independent. This result is in line with what obtained in investigations of the adsorption kinetics via ellipsometry[16], and explained in terms of the direct correlation between the saturating value of the adsorbed amount after long annealing time and the interfacial potential[36]. As the latter quantity corresponds to energies on the order of ~1$k_BT$[1,33], $\Delta H_{des}$ is not expected to exceed experimental uncertainties within the 20 K covered by our experiments. Secondly, an analysis of the temperature dependence of time needed by $\Delta H_{des}$ to reach saturation, see inset of Fig. 4, permitted us to identify the molecular origin of this process. Fits of the experimental data to the Arrhenius equation, $\tau_{eq} = \tau_0 \exp(E_a/k_BT)$, provided an activation energy $E_a$ of ~340 kJ mol$^{-1}$. Such value, too high for simple thermally activated processes, is common in the case of molecular rearrangements requiring cooperativity[37]. Hence, we tested[38] if the temperature dependence of $\tau_{eq}$ could be associated with the segmental $\alpha$-relaxation of P$t$BS[24], that is, the cooperative relaxation mechanism responsible for density fluctuations, arising from changes in molecular conformations involving several structural units[39]. This analysis confirmed that adsorption has an activation energy comparable to that of segmental mobility, although the two processes are separated in time by almost 6 orders in magnitude, $\tau_{eq} \approx 10^{5.7}\tau_\alpha$. Such a trend is in line with similar observations for films of PS and other PS analogs[8,40]. Measurements based on the heat released upon desorption permitted us to verify that adsorption is an ultraslow process coupled to density fluctuations, and that addition of new monomers to the adsorbed layer is possible only after a tremendously large number of molecular rearrangements.

Finally, we remark that adsorption was not observed when cooling the system with the same rate used for heating. This phenomenon hints at a possible hysteresis as in the case of first-order wetting transitions[41,42]. We anticipate that such a feature could be exploited to tune the density of a polymer coating upon controlled desorption, e.g., by heating the thin layer for a time shorter than what required to fully desorb the whole set of interfacial chains.

In conclusion, we have demonstrated that it is possible to investigate, by FSC, the desorption of a polymer melt from a solid substrate. We performed such experiments on thin layers of P$t$BS and PS previously adsorbed on silicon oxide. Though tremendously relevant for the performance and stability of thin polymer coatings, the desorption of polymer melts upon heating was not previously characterized because of concurrent phenomena of chemical degradation. Our investigation permitted to clarify that desorption can be treated as a first-order phase transition, in ways analogous to melting. Based on the energetic analysis of the detachment of polymer chains from a melt, we conclude that most of the enthalpy of desorption is related to the conformational changes occurring at this liquid/liquid phase transition. We are confident that our work will stimulate fruitful discussion and inspire the development of a solid theoretical framework assessing the impact of chain architecture and connectivity on adsorption/desorption of polymer melts.

## Methods

**Materials.** Poly(4-*tert*-butylstyrene) (P*t*BS) with number average molecular weight $M_n = 48$ kg mol$^{-1}$ and polydispersity index PDI = 2.7 was purchased from Aldrich and used as received. Films with thickness ranging from 40 and 80 nm were prepared by spin-coating solutions of P*t*BS in toluene (Aldrich, purity 99.9 %) directly on the backside of chips for FSC, which is made by silicon oxide[43]. A similar procedure was followed for the preparation of polystyrene (PS, $M_n = 88$ kg mol$^{-1}$, PDI = 1.07, from Polymer Source Inc.) thin films. In this case, film thicknesses were ranging between 60 and 90 nm. The sample thickness was determined via conventional methods considering the specific heat step at the glass transition, as described elsewhere[43].

**Calorimetry measurements.** FSC was carried by the Flash DSC 1 by Mettler Toledo based on chip calorimetry technology, equipped with a two-stage intra-cooler, allowing for temperature control between 183 and 773 K. The FSC cell was constantly fluxed by nitrogen gas with rate of 20 ml min$^{-1}$. Calibration was ensured through the melting of a standard indium at different rates placed onto the reference chip-area afterward. Samples were first heated at 743 K for 10 s. Such a preliminary procedure aims to erase the thermo-mechanical history induced by spin-coating. This marks an important difference with previous adsorption experiments, where heating was limited to $T_g$ +60 K, which is not enough to erase conformational peculiarities induced by spin-coating[44]. After this preliminary procedure, samples were cooled down at 4000 K s$^{-1}$ and immediately reheated with $10^4$ K s$^{-1}$ up to 743 K to collect a reference scan. This was followed by a standard procedure[45] of annealing between 443 and 473 K, in steps of 5 K, to adsorb P*t*BS on the substrate. This corresponds to $T \geq T_g$ +45 K, where $T_g$ ~398 K is the glass transition temperature obtained at ~10 K min$^{-1}$, which allows attaining significant adsorption within hours[9]. After that, samples were cooled at 4000 K s$^{-1}$ to room temperature and immediately heated up at $10^4$ K s$^{-1}$ to 743 K for data recording. Finally, an additional heating scan at the same rate was recorded after cooling to room temperature at 4000 K s$^{-1}$.

## Data availability

The data sets generated during and/or analyzed during the current study are available from the corresponding authors on reasonable request.

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

## Acknowledgements

We thank Sanat K. Kumar and Luis G. MacDowell for a fruitful discussion on the physics of adsorption. D.C. acknowledges financial support from the project PGC2018-094548-B-I00 (MICINN-Spain and FEDER–UE) and the project IT-1175-19 (Basque Government). S.N. acknowledges financial supports from the Action Concertée Recherche-ULB under project SADI and the Fonds de la Recherche Scientifique FNRS under Grant EXOTICAGE.

## Author contributions

S.N. and D.C. conceived the idea and supervised the project. X.M. performed the experiments and analyzed the data. X.M., S.N. and D.C. contributed equally to the discussion and the redaction of the manuscript.

## Competing interests

The authors declare no competing interests.
