## [Peer Review File · Nature Communications]

REVIEWER COMMENTS

Reviewer #1 (Remarks to the Author):

This manuscript reports the first direct measurement of desorption thermodynamics of a polymer from a solid substrate. The temperature of desorption can be a high value. The value reported in this manuscript is $\sim 660\text{K}$. As a result, assessing desorption thermodynamics, i.e., enthalpy of desorption, is challenging as material degradation can occur. To overcome this challenge, the authors used fast scanning calorimetry to "by pass" degradation to measure desorption. The results systematically show how annealing time impacts the desorption enthalpy. The manuscript presents a well articulated story of novel experiments. The manuscript should be published in Nature Communications. I have two additional comments:

1. I found the comparison (analogy) to crystallization not necessary. For instance, the manuscript estimates T_m for PtBS even though it does not crystallize, as the authors pointed out. (Also from calorimetry it is confirmed that the desorption observed in this manuscript is first order.)
2. On page 12, the manuscript cites reference 18 in the context of changing solvent quality. In ref 18, only one solvent was used. Hence, the quality of the solvent was unchanged.

Reviewer #2 (Remarks to the Author):

The authors propose the new thermodynamic concept about polymer desorption. They utilize the novel approach, fast scanning calorimetry (FSC), and studied the adsorption/desorption behavior of poly(4-tert-butylstyrene) (PtBS) chains deposited on the solid coated with a silicon oxide layer. The results are informative and the discussions behind the phenomenon are thoughtful. I believe the work offers an important aspect in the field. However, my major concern is that the authors used only one polymer with a fixed molecular weight, one heating rate, and one substrate to draw the conclusion: polymer adsorption/desorption is the first-order phase transition. It is known that results obtained with calorimetric methods strongly depend on heating rates. In fact, according to their data shown in Fig. 1a, the T_g value obtained from the FSC is about 30C higher than the bulk. So the question is whether the transition they claim is inherent. Can the authors show systematic FSC data with different heating rates to validate the statement? Also, I understand the choice of PtBS as the model, but polystyrene (PS) would be a better system since the degradation temperature is much lower than PtBS, monodisperse PS with different molecular weights are cheaply available commercially, and the mechanism of polymer adsorption has been extensively studied by the authors and other groups. In addition, depending on the interactions between a polymer and substrate as well as chain flexibility, the conformations of an adsorbed polymer chain and the desorption mechanism should be different. Moreover, the comparison between polymer adsorption/desorption and crystallization/melting with PtBS (on Page 10) is not convincing. For that purpose, it would be interesting to use semicrystalline polymers, such as PET where the adsorption behavior is well characterized (Macromolecules 2017, 50, 6804) or syndiotactic (or isotactic) PS, and to study the melting/crystallization transition as well as adsorption/desorption transition with FCS. Without such additional experimental supports/evidence, their conclusion may not be justified and the manuscript may not be fully extended yet to merit a publication in the journal.

Reviewer #3 (Remarks to the Author):

Referee report

on **First direct observation of desorption of a melt of long polymer chains**

by Xavier Monnier, Simone Napolitano & Daniele Cangialosi

The authors report on a direct observation (using fast scanning calorimetry) of a desorption transition for a polymer melt from a solid wall. Such a thermodynamic transition is very difficult to observe as normally the polymer chains are degraded before the transition occurs. Fast scanning calorimetry allows the measurement to be faster than degradation. Overall, the manuscript is well written, the realisation of the experiment and its analysis are well explained, and the results are interesting. Explanations and conclusions are instructive. I have a number of individual points that should be changed to make the manuscript more accessible for a readership that consists of experimentalists and theoreticians in the wider field of soft matter.

Then I believe, the manuscript can be published.

1. Title: Many journals prefer not to have self-assessments as “first” or “novel” in a text. The referee is of the same opinion. If a “direct observation” is novel then the informed reader will recognise and value it, and for the non-informed reader it does not matter. So dropping the “first” does not diminish the impact, but spares the authors the embarrassment if it is later found that someone else had done something similar before.
2. Intro, p.2: I understand sentence 1, 2 and 4, but the role of “The related changes in particle density in proximity of the transition zone between two or more phases determines the stability of the whole system.” is not clear. Please explain better what is meant. Does “stability” here relate to instability in relation to possible pattern formation?
3. Intro, p.2, final sentence of paragr.2: something is amiss in the sentence.
4. p.3, l.4: It is stated that the argument against desorption is based on kinetics. But isn't the kinetic argument itself at the core an energetic one, as ΔE determines the probability via a Boltzmann factor?
5. On pg. 3, adsorption is nicely introduced as a phase transition depending on control parameters concentration and temperature. It would be instructive to give a phase diagram in these parameters and sketch what are the paths taken by sample preparation, annealing and during the fast scanning. It would be, in particular, useful to discuss in this context which preparation steps are actually nonequilibrium processes and which are phase transitions in the 'quasi-equilibrium' sense.
6. pg.5: If the mentioned glass transition mentioned (at 433K) is visible in Fig.1b, it should be mentioned how, and if not this also should be mentioned and briefly explained. As the run passes through the 433K some readers could expect to see it there.
7. p.5, bottom: g_{sample} should be explained.

8. Fig.1: be kind to the casual reader: Mention in the caption what is actually seen on the y-axis, add ticks and tick labels (there are none at all) - also for the inset. Is the inset of (b) at a particular temperature? What means "Endo up"? The curious lines going to the inset are not needed, barely visible, confusing and should be dropped as they are not needed if plot and inset have proper axes ticks and labels.
9. Fig.2: same as Fig.1
10. p.10, l2: "ustmost"?
11. As on p.10 crystallization/melting and adsorption/desorption are classified as first order transitions, the question arises how important hysteresis effects are in the performed experiments. Please add a short comment.
12. Fig.4: The scheme is overall fine, but the rationale behind the various colourings of the beads remains elusive to me. If there is a rationale, please explain. If there is none, make it less confusing.
E.g., why when melting, a single long chain breaks in 6 different ones; or why does a desorbing chain become shorter.
13. p.12, bottom: why is "varying the quality of the solvent" termed "artificial" but fast heating not? Are these not only experiments using different control parameters? Why is one "better" than the other?
14. p.13, l.2: what is meant by "way far from"?; further down on line 16 "desorption" is misspelled.
15. Intro and conclusion: I stumbled over statements that seem to contradict each other: On the one hand it is said that degradation normally preempts desorption, therefore fast scanning calorimetry is needed. On the other hand, it is emphasised in the introduction and conclusion that desorption can be used to "finely tune the performance of nanomaterials" and is "tremendously relevant for the performance and stability of thin polymer coatings", respectively. Somehow, these statements seem to contradict each other. Would not degradation be the practically more important issue?
16. Above critique of figures also applies to figures in the Supplement.
17. Final point: It becomes increasingly good practise to provide the community and wider society with the original data. Will the original data for the figures be available on a data repository like zenodo or figshare (even if still embargoed)? Then a remark with a doi link would be helpful.

Reviewer #1:

This manuscript reports the first direct measurement of desorption thermodynamics of a polymer from a solid substrate. The temperature of desorption can be a high value. The value reported in this manuscript is ~ 660K. As a result, assessing desorption thermodynamics, i.e., enthalpy of desorption, is challenging as material degradation can occur. To overcome this challenge, the authors used fast scanning calorimetry to "by pass" degradation to measure desorption. The results systematically show how annealing time impacts the desorption enthalpy. The manuscript presents a well articulated story of novel experiments. The manuscript should be published in Nature Communications. I have two additional comments:

Response: We thank this Reviewer for the positive assessment on our manuscript and for recommending it for publication in Nature Communications. In the following, we reply to the point raised:

Reviewer's comment: 1. I found the comparison (analogy) to crystallization not necessary. For instance, the manuscript estimates T_m for PTBS even though it does not crystallize, as the authors pointed out. (Also from calorimetry is it confirmed that the desorption observed in this manuscript is first order.)

Response: We agree with this Reviewer that, as PTBS does not crystallize, the comparison between desorption and melting temperature cannot be done. Furthermore, this point was also raised by Reviewer 2. Hence, we have followed both Reviewers' comments and skipped this comparison. In the revised version of the manuscript, we simply highlight that the large temperature of desorption can be understood considering the mild difference between the entropy of the adsorbed and desorbed polymer.

Reviewer's comment: 2. On page 12, the manuscript cites reference 18 in the context of changing solvent quality. In ref 18, only one solvent was used. Hence, the quality of the solvent was unchanged.

Response: Indeed in ref. 18 only one solvent was used. However, in that reference the solvent volume fraction was changed. Hence, we have specified this in the revised manuscript: "...where a reduction in adsorbed amount was obtained in artificial conditions by varying the quality **or the volume fraction** of the solvent" (page 14, line 6).

Reviewer #2:

Reviewer's comment: The authors propose the new thermodynamic concept about polymer desorption. They utilize the novel approach, fast scanning calorimetry (FSC), and studied the adsorption/desorption behavior of poly(4-tert-butylstyrene) (PtBS) chains deposited on the solid coated with a silicon oxide layer. The results are informative and the discussions behind the phenomenon are thoughtful. I believe the work offers an important aspect in the field. However, my major concern is that the authors used only one polymer with a fixed molecular weight, one heating rate, and one substrate to draw the conclusion: polymer adsorption/desorption is the first-order phase transition. It is known that results obtained with calorimetric methods strongly depend on heating rates. In fact, according to their data shown in Fig. 1a, the T_g value obtained from the FSC is about 30C higher than the bulk. So the question is whether the transition they claim is inherent. Can the authors show systematic FSC data with different heating rates to validate the statement? Also, I understand the choice of PtBS as the model, but polystyrene (PS) would be a better system since the degradation temperature is much lower than PtBS, monodisperse PS with different molecular weights are cheaply available commercially, and the mechanism of polymer adsorption has been extensively studied by the authors and other groups. In addition, depending on the interactions between a polymer and substrate as well as chain flexibility, the conformations of an adsorbed polymer chain and the desorption mechanism should be different.

Response: The Reviewer points out that, to make our arguments more convincing, we should extend our measurements to other polymer/substrate systems. Hence, as suggested, we have decided to pursue a significant effort to complement results on PTBS with those on the archetypal polymer used in thin films studies, PS. In this case, due to the limited thermal stability, the production of a fresh PS film was required for the determination of the desorption behavior after annealing at each temperature.

Our results on this polymer, reported in the Supplementary Information, show the following outcome: 1) In ways analogue to PTBS (whose *mechanisms of adsorption have also been extensively studied by the authors*), PS also exhibits a desorption peak in a temperature range much higher than T_g; 2) The desorption temperature is independent of the previously chosen deposition temperature. Both results are in line with our first order phase transition interpretation. Apart from the presentation of these results in the Supplementary Information, we have discussed them in the main manuscript (page 8, first 6 lines).

Regarding the effect of the heating rate, even in the case of melting, archetypal first order thermodynamic transition, a heating rate dependence (of the transition temperature) is usually observed, because of superheating effects. This is especially evident at the high heating rates typical of fast scanning calorimetry, as deeply investigated by Toda and co-workers; see for instance: *Thermochimica Acta* 677, 211-216, 2019. Our experiments can be carried out only over a limited heating rate range due incipient degradation and depression of heat flow signal at low rates. Hence, changing over a limited range the heating rate would not provide any additional information.

Changing the substrate would certainly be an aspect deserving further investigation. However, in our work we directly deposit our samples on the chip for fast scanning

calorimetry. A modification of the nature of the substrate would require major technical developments, which are beyond the scope of our study.

Finally, we remark that our study aims to prove that adsorption/desorption exhibit all the features of a first order thermodynamic transition. This was done presenting clear results on PTBS and now also on PS. Hence, the purpose of our study was not that of investigating the effect of chain connectivity on the desorption behavior, by changing the polymer molecular weight. This kind of work would require 1) a solid theoretical framework, not present at the state of art; - furthermore, because of the extremely long computational times necessary to observe desorption of polymer chains, *in silico* simulations are actually still limited to oligomers, see J. Chem. Phys. 146, 203308 (2017); 2) *ad-hoc* synthesized samples of identical chemistry (synthesis, presence of contaminants). Employing commercial samples, for which the synthesis details are often unknown, would not yield reliable results, because adsorption is extremely sensitive to contaminants and surface preparation. In our work, we followed a rigorous protocol, which permitted to obtain reproducible results. We hope that our work on desorption will stimulate the discussion among theorists and the design of dedicated experiments, so that we, or other colleagues, will be able to address this problem in the future.

Reviewer's comment: Moreover, the comparison between polymer adsorption/desorption and crystallization/melting with PtBS (on Page 10) is not convincing. For that purpose, it would be interesting to use semicrystalline polymers, such as PET where the adsorption behavior is well characterized (Macromolecules 2017, 50, 6804) or syndiotactic (or isotactic) PS, and to study the melting/crystallization transition as well as adsorption/desorption transition with FCS. Without such additional experimental supports/evidence, their conclusion may not be justified and the manuscript may not be fully extended yet to merit a publication in the journal.

Response: We agree with the Reviewer and therefore, also in view of Reviewer 1 comment, we have decided to leave out the comparison between melting and desorption temperature. We remark that the use of non-crystalline polymers (PTBS and now PS) is preferred in this type of studies. The richer set of transitions taking place in semi-crystalline polymers could mask, or even convolute with, the adsorption/desorption peaks. This would complicate the analysis, and the interpretation of the results would not be as straightforward as in the case of non-crystalline polymers (e.g. PTBS and PS).

Reviewer #3:

The authors report on a direct observation (using fast scanning calorimetry) of a desorption transition for a polymer melt from a solid wall. Such a thermodynamic transition is very difficult to observe as normally the polymer chains are degraded before the transition occurs. Fast scanning calorimetry allows the measurement to be faster than degradation. Overall, the manuscript is well written, the realisation of the experiment and its analysis are well explained, and the results are interesting. Explanations and conclusions are instructive. I have a number of individual points that should be changed to make the manuscript more accessible for a readership that consists of experimentalists and theoreticians in the wider field of soft matter. Then I believe, the manuscript can be published.

Response: We thank this reviewer for the positive assessment on our manuscript and for recommending it for publication in Nature Communications. In the following, we reply to the points raised.

Reviewer's comment: 1. Title: Many journals prefer not to have self-assessments as "first" or "novel" in a text. The referee is of the same opinion. If a "direct observation" is novel then the informed reader will recognise and value it, and for the non-informed reader it does not matter. So dropping the "first" does not diminish the impact, but spares the authors the embarrassment if it is later found that someone else had done something similar before.

Response: We agree with the reviewer, the new title of our manuscript now reads "Direct observation of..". Furthermore we have also removed the word "first" in the abstract, where we claim that our finding is a clear-cut evidence of desorption.

Reviewer's comment: 2. Intro, p.2: I understand sentence 1, 2 and 4, but the role of "The related changes in particle density in proximity of the transition zone between two or more phases determines the stability of the whole system." is not clear. Please explain better what is meant. Does "stability" here relate to instability in relation to possible pattern formation?

Response: We recognize the ambiguity of the previous sentence, the text now reads: "The related changes in particle density in proximity of the transition zone between two or more phases determines the **performance** of the whole system".

Reviewer's comment: 3. Intro, p.2, final sentence of paragr.2: something is amiss in the sentence.

Response: We carefully checked the sentence indicated by the reviewer, and modified it. Now the sentence reads: "Being able to control the latter quantity (e.g. by simple thermal treatments applicable at the industrial scale) would open the way to new processing strategies to finely tune the interfacial density and, hence, the properties of nanomaterials."

Reviewer's comment: 4. p.3, l.4: It is stated that the argument against desorption is based on kinetics. But isn't the kinetic argument itself at the core an energetic one, as E determines the probability via a Boltzmann factor?

Response: The reviewer is correct; kinetics and energy are intimately correlated. Based on this correlation, in the temperatures and at the timescales of technological interest, the energetic contribution is rather large, which provides a very small probability of desorption. The manuscript has been modified in line with this comment (page 3, lines 4-5).

Reviewer's comment: 5. On pg. 3, adsorption is nicely introduced as a phase transition depending on control parameters concentration and temperature. It would be instructive to give a phase diagram in these parameters and sketch what are the paths taken by sample preparation, annealing and during the fast scanning. It would be, in particular, useful to discuss in this context which preparation steps are actually nonequilibrium processes and which are phase transitions in the 'quasi-equilibrium' sense.

Response: This is a great idea! We followed the recommendation of the reviewer and added a new figure, now labeled as Figure 1, where we show both the thermodynamics and kinetics (via a TTT diagram) of the temperature and time protocol applied in this study. We are convinced that this has tremendously boosted the readability of our manuscript.

Reviewer's comment: 6. pg.5: If the mentioned glass transition mentioned (at 433K) is visible in Fig.1b, it should be mentioned how, and if not this also should be mentioned and briefly explained. As the run passes through the 433K some readers could expect to see it there.

Response: While in the text we indicated that the glass transition corresponds to a step in C_p , in the figure (now labeled as 2b) we now explicitly point at the glass transition temperature.

Reviewer's comment: 7. p.5, bottom: m_{sample} should be explained.

Response: We were referring to the mass of the sample in grams; we now use a more direct symbol, m_{sample} .

Reviewer's comment: 8. Fig.1: be kind to the casual reader: Mention in the caption what is actually seen on the y-axis, add ticks and tick labels (there are none at all) - also for the inset. Is the inset of (b) at a particular temperature? What means "Endo up"? The curious lines going to the inset are not needed, barely visible, confusing and should be dropped as they are not needed if plot and inset have proper axes ticks and labels.

Reviewer's comment: 9. Fig.2: same as Fig.1

Response: We have modified both figures according to the reviewer's comments.

Reviewer's comment: 10. p.10, l2: "ustmost"?

Response: This typo has been corrected.

Reviewer's comment: 11. As on p.10 crystallization/melting and adsorption/desorption are classified as firstorder transitions, the question arises how important hysteresis effects are in theperformed experiments. Please add a short comment.

Response: The Reviewer here raises a very good and pertinent point. Adsorption was not observed when cooling the system with the same rate used for heating, hinting at a possible hysteresis, as observed in the case of first-order wetting transitions. We are currently collaborating with some theoreticians to understand whether nucleation could be at the origin of these hysteresis effects. We anticipate that such task will be achieved by extending current theories, based on vapor pressure, to advanced models which could reproduce our experiments, where the control parameter is the temperature. While it is premature to inform the readers on these details, in the text we added the following paragraph: "Finally, we remark that adsorption was not observed when cooling the system with the same rate used for heating. This phenomenon hints at a possible hysteresis as in the case of first-order wetting transitions. We anticipate that such feature could be exploited to tune density of a polymer coating upon controlled desorption, e.g. by heating the thin layer for a time shorter than what required to fully desorb the whole set of interfacial chains" (page 16, line 3-7).

Reviewer's comment: 12. Fig.4: The scheme is overall fine, but the rational behind the various colourings of the beads remains elusive to me. If there is a rational, please explain. If there is none, make it less confusing. E.g., why when melting, a single long chain breaks in 6 different ones; or why does a desorbing chain become shorter.

Response: We have modified the caption, which now clearly indicates that non-adsorbed chains are colored in pastel tones of blue, while adsorbed chains are in tones of green and directly adsorbed monomers in dark green. While in real crystals, except in proximity of defects, it would be difficult to identify different chains, in the new version of this Figure (now labeled as Figure 5) we now use different tones of green to ease the difference. Finally, we have corrected the issue with the desorbing chain.

Reviewer's comment: 13. p.12, bottom: why is "varying the quality of the solvent" termed "artificial" but fast heating not? Are these not only experiments using different control parameters? Why is one "better" than the other?

Response: Diluting with a solvent implies adding a different type of molecules to the system, a condition which is not required when a pure system (polymer only) is heated.

Reviewer's comment: 14. p.13, l.2: what is meant by "way far from"?

Response: Same as above, when a solvent is added, the system is no longer composed by the same type of molecules. Desorption in such case is achieved because of entropy, due to the larger ensemble of conformations a chain could experience in the solvent. We can also discuss the same phenomenon in terms of density: by adding a large solvent reservoir in the system, we push the density of chains to values virtually null, while the density of a melt (thermal expansion coefficient $\sim 10^{-3} \text{ K}^{-1}$) is not particularly affected by temperature. We modified our text by better highlighting the difference between the two cases (page 14, lines 9-10).

Reviewer's comment: further down on line 16 "desorption" is misspelled.

Response: We corrected the misspelled term in two places of the manuscript.

Reviewer's comment: 15. Intro and conclusion: I stumbled over statements that seem to contradict each other: On the one hand it is said that degradation normally preempts desorption, therefore fast scanning calorimetry is needed. On the other hand, it is emphasised in the introduction and conclusion that desorption can be used to "finely tune the performance of nanomaterials" and is "tremendously relevant for the performance and stability of thin polymer coatings", respectively. Somehow, these statements seem to contradict each other. Would not degradation be the practically more important issue?

Response: Here we are referring to protocols that can be implemented at industrial level, where an adsorbed sample (e.g. a sample held for enough time above T_g , during a previous processing step) would undergo a fast thermal treatment allowing for partial desorption. The reviewer is right: this treatment should be fast enough to avoid degradation. We modified the conclusions to better clarify this point (page 16, last 3 lines).

Reviewer's comment: 16. Above critique of figures also applies to figures in the Supplement.

Response: We have modified those figures according to the reviewer's comments.

Reviewer's comment: 17. Final point: It becomes increasingly good practise to provide the community and wider society with the original data. Will the original data for the figures be available on a data repository like zenodo or figshare (even if still embargoed)? Then a remark with a doi link would be helpful.

Response: On request of the readers of our manuscript, following the guidelines of the journal, we will be keen to provide the community with original data.

REVIEWERS' COMMENTS:

Reviewer #1 (Remarks to the Author):

The authors have adequately addressed the comments of all three referees, and have revised their manuscript accordingly. I recommend publication in Nature Communications.

Reviewer #2 (Remarks to the Author):

The authors have carefully and thoughtfully addressed the concerns raised by the reviewer to highlight the novel characteristic of the polymer adsorption/desorption process on a solid. I am happy to recommend the revised manuscript for publication in the journal.

Reviewer #3 (Remarks to the Author):

> The authors have amended the manuscript following most of my points, e.g., have introduced the instructive scheme in Fig. 1 and increased clarity of other figures. In my opinion the manuscript can now be published.

> However, I have a comment regarding the data availability that you should forward to the authors:

> It is now stated that ``On request of the readers of our manuscript, following the guidelines of the journal, we will be keen to provide the community with original data.' This implies that one has to trust the institution of the authors to guarantee long-term data storage and access. My experience says that, in practice, this seldomly works and public data repositories are a much better choice. I can not see good reasons not to do this besides that one shies away from the additional work of bringing the data in a format others can understand.